# Consistent 3D Object Detection with Active LLM Reasoning

## Abstract

Maintaining semantic label consistency across multiple views is a persistent challenge in 3D semantic object detection. Existing zero-shot approaches that combine 2D detections with vision-language features often suffer from bias toward non-descriptive viewpoints and require a fixed label list to operate on. We propose a truly open-vocabulary algorithm that uses large language model (LLM) reasoning to relabel multi-view detections, mitigating errors from poor, ambiguous viewpoints and occlusions. Our method actively samples informative views based on feature diversity and uncertainty, generates new label hypotheses via LLM reasoning, and recomputes confidences to build a spatial-semantic representation of objects. Experiments on controlled single-object and diverse multi-object scenes show over 40% improvement, in accuracy and sampling rate over ubiquitous fusion methods using YOLO, and CLIP. We demonstrate in multiple cases that our **LLM**-guided **A**ctive **D**etection and **R**easoning (LADR) balances detail preservation with reduced ambiguity and a low sampling rate.

## 1 Introduction

Consistently detecting objects across multiple viewpoints is a crucial task for autonomous agents, such as drones and robots. A single object may appear vastly different depending on the viewpoint, lighting, or degree of occlusion, and visual features extracted from such views often drift in embedding space. As a result, inconsistent labels emerge when fusing detections across views, leading to degraded spatial-semantic representations and downstream performance.

Recent zero-shot approaches (Jatavallabhula et al., 2023; Peng et al., 2023; Cartillier et al., 2021), address this by combining off-the-shelf detectors (Redmon et al., 2016) with vision-language models (Radford et al., 2021; Cherti et al., 2023) to assign open-vocabulary labels in 3D. While these methods avoid task-specific retraining, they rely heavily on two components: (1) the accuracy of the underlying detector, and (2) the similarity between extracted image features and a user-defined list of candidate labels. Both dependencies introduce bottlenecks. First, misdetections or low-quality views (such as those from the back of an object) can dominate the fused feature representation, biasing the final label. Second, reliance on a user-defined list of labels limits true open-vocabulary capability, hampers generalization to novel categories, and constrains the level of detail that can be captured for each object.

We propose a different approach referred to as **LADR** (**LLM**-guided **A**ctive **D**etection and **R**easoning). LADR uses large language model (LLM) reasoning to actively refine and reweight multi-view detections. Instead of passively aggregating features, our method iteratively samples informative viewpoints based on feature diversity, prompts an LLM to generate and refine label hypotheses from available visual evidence, and recomputes label confidences accordingly. This reasoning process reduces the influence of misleading views, removes the need for a fixed label set, and enables a more robust spatial-semantic representation of the scene.

Our contributions are as follows:

- **LLM-guided relabeling for 3D consistency:** An open-vocabulary method that uses LLM reasoning to correct viewpoint-induced misclassifications without retraining.
- **Smart sampling strategy:** An active selection of views based on feature diversity, balancing detail preservation with reduced context ambiguity, and lower sampling rate.

- **Spatial-semantic mapping:** A representation that integrates refined labels with object geometry, suitable for downstream 3D tasks.
- **Comprehensive evaluation:** single-object experiments, and multi-object scene experiments across diverse environments, showing improvement of over 40%, respectively, in 3D semantic label accuracy and sampling rate, over ubiquitous fusion methods using YOLO, and CLIP.

Our contributions establish a framework for open-vocabulary 3D understanding that combines semantic reasoning, efficient view selection, and spatial integration, leading to more robust and consistent labeling across diverse scenarios.

## 2 RELATED WORK

### 2.1 FOUNDATION MODELS IN OBJECT DETECTION

Object detection has rapidly advanced from region-based CNNs and single-stage detectors to foundation models, which enable more general and flexible representations beyond closed-set training. Architectures such as YOLO-World and YOLOE (Cheng et al., 2024; Wang et al., 2025) leverage large-scale pretraining to improve detection accuracy and adaptability across diverse scenarios. Vision-language models (VLMs) like CLIP (Radford et al., 2021; Cherti et al., 2023) provide openvocabulary capabilities by connecting visual features with text embeddings, while models such as Segment Anything (Kirillov et al., 2023) offer class-agnostic segmentation that can be integrated into detection pipelines. Multimodal large language models like GPT-4V (OpenAI, 2024) further complement these approaches by enabling zero-shot reasoning over visual inputs, making them useful for refining labels and guiding exploration. These approaches demonstrate the potential to reduce reliance on task-specific training and expand detection to previously unseen categories.

### 2.2 OPEN-VOCABULARY 3D OBJECT DETECTION

ConceptFusion (Jatavallabhula et al., 2023) builds open-vocabulary 3D object maps by combining pretrained VLMs with 3D scene representations. The method uses YOLO-World as an initial object detector and Segment Anything for segmentation, attaching VLM features (e.g., from CLIP) to 3D points reconstructed from RGB-D scans, with features from multiple 2D observations aggregated via simple averaging (which ignores the 3D consistency problem). While it aims to assign openvocabulary labels, the object categories are ultimately constrained to a fixed set. Peng et al. (2023) takes a voxel-based approach, backprojecting per-pixel CLIP features into a 3D voxel grid and fusing multiple views using different pooling strategies (random, median, or mean) among these approaches, mean pooling yields the most stable results. Kassab et al. (2024) revisits design choices for open-vocabulary 3D labeling by selecting a single "best" view per object based on a confidence metric, with the entropy of CLIP similarities with category embeddings performing best. In contrast, LADR leverages LLM reasoning to iteratively identify and reweight informative views, producing a more robust spatial-semantic representation that is less sensitive to viewpoint bias and not limited by a fixed label set.

### 2.3 ACTIVE EXPLORATION

Active exploration in embodied agents aims to optimize camera or agent trajectories to reduce uncertainty and collect informative observations. SEAL Chaplot et al. (2021) and subsequent works Scarpellini et al. (2024) introduce a self-supervised framework in which agents explore their environment to learn semantic segmentation without manual labels, leveraging 3D spatial consistency. These methods train an exploration policy to target novel or uncertain areas, optimizing coverage of diverse object views. Features from multiple viewpoints are reprojected into a shared 3D space using depth and camera poses, and a 3D consistency loss ensures that features corresponding to the same physical point remain consistent across views. This supervision enables learning of a semantic segmentation function directly from RGB-D frames, without human annotations, and replaces random or fixed path planning with informed, targeted exploration. While effective, these approaches often require reinforcement learning policies and multiple rollouts, which can be computationally expensive. In contrast, zero-shot LLM-based methods can reason about object semantics

directly from observations without task-specific policy training, avoiding the overhead and sample inefficiency inherent to learned exploration strategies.

## 3 THE 3D CONSISTENCY PROBLEM

Achieving consistent object labeling across multiple viewpoints remains a key obstacle in 3D perception. In multi-view pipelines, each observation of an object is processed independently before being fused into a unified label. When these observations are heterogeneous (due to varying viewpoints, occlusions, or lighting) the resulting feature embeddings can drift toward non-representative appearances. This drift can overweight misleading views, leading to label instability.

In zero-shot approaches such as those combining YOLO detections with CLIP embeddings, the problem is exacerbated by two factors:

1. **Viewpoint sensitivity:** Descriptive views (e.g., the front of a piano) and non-descriptive views (e.g., the back of the same piano) contribute equally to the aggregated embedding. If the majority of views lack discriminative features, the resulting label can shift toward incorrect categories.

2. **Label space constraints:** Even in open-vocabulary settings, relying on a fixed set of candidate labels constrains the level of detail that can be captured for each object, e.g., labeling a chair simply as 'furniture' rather than distinguishing it as an 'office swivel chair.'

To illustrate the severity of this issue, we consider a controlled example where images are taken around a piano. We define **good views** as those from the front, containing distinctive features, and **bad views** as those from the back, lacking such cues. In a progressive experiment, we start with three good views and incrementally replace them with bad ones, testing multiple labeling strategies. The task is to assign a single label to the object, given all current views.

| Method | 3 Good / 0 Bad | 2 Good / 1 Bad | 1 Good / 2 Bad | |
|---|---|---|---|---|
| YOLOE Constrained | **piano (0.25)** | **piano (0.21)** | crate (0.26) | 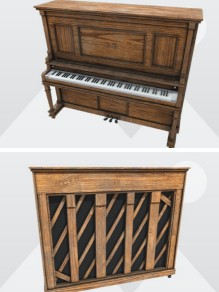 |
| YOLOE ScanNet200 | cabinet (0.78) | cabinet (0.61) | cabinet (0.51) | |
| YOLOE RAM | chiffonier (0.90) | wall (0.16) | wall (0.17) | |
| CLIP Constrained | **piano (0.31)** | **piano (0.27)** | crate (0.26) | |
| CLIP ScanNet200 | **piano (0.31)** | **piano (0.27)** | crate (0.26) | |
| CLIP RAM | **piano (0.31)** | **piano (0.27)** | oak (0.27) | |
| LLM | **acoustic piano** | **acoustic piano** | **acoustic piano** | |

Table 1: **Piano viewpoint bias experiment.** "Good" images show the piano front, while "Bad" images show the back. Each cell reports the *predicted label (confidence)*, with correct predictions shown in **bold**. For YOLOE baselines, the most frequent label is selected, whereas CLIP baselines choose the label with the highest similarity. The "Constrained" setting restricts candidate labels to "piano" and "crate," while "ScanNet200" (Rozenberszki et al., 2022) and "RAM" (Recognize Anything Model class list of over four thousand categories, (Zhang et al., 2023)) use their respective class lists to select the most probable label. The LLM is prompted to give a more specific label than a simple class label."

As shown in Table 1, methods relying solely on YOLO or CLIP degrade quickly as bad views increase. In the 1-good / 2-bad case, CLIP-based methods incorrectly label the piano as "crate" or "oak," while YOLO struggles even more, particularly when the label space is large, producing highly inconsistent predictions. In contrast, the LLM-based approach consistently selects the correct and more detailed label across all conditions. However, the LLM does not provide calibrated confidence values, making it difficult to assess the reliability of its predictions on its own. This observation motivates LADR's hybrid strategy: combining the reasoning capabilities of LLMs with the quantitative confidence scores from CLIP allows for both robust label selection and informed weighting across views, mitigating the effects of viewpoint bias and constrained label spaces.

## 4 NOTATION AND WORKFLOW

We consider multi-view object labeling in 3D scenes. For simplicity, and without loss of generality, we address a single object. Let $\mathcal{I} = \{I_1, \ldots, I_N\}$ denote the initial set of RGB-D images captured around a target object, where $N$ is the number of views. Each image $I_i$ is accompanied by depth information $D_i$ and camera pose $P_i$. For each image, object observations are extracted using a combination of a detector, a feature extractor and a segmentation model as

$$O_i = \text{DetectAndSegment}(I_i, D_i, P_i),$$

and merged across all views into a spatial-semantic map

$$\mathcal{M} = \text{MergeObservations}(\{O_1, \ldots, O_N\}),$$

which accumulates object points, labels, and features into a coherent 3D representation, analogous to ConceptFusion's fusion (Jatavallabhula et al., 2023). We define a function for relabeling as

$$\mathcal{M}_{\text{refined}}, \ P_{\text{next}} = \text{RefineAndPropose}(\mathcal{M}, \mathcal{I}),$$

which applies LLM reasoning to refine labels in $\mathcal{M}$ and selects the most informative next viewpoint.

Our workflow proceeds iteratively: images are captured and merged into $\mathcal{M}$, refined, and used to propose the next viewpoint. This repeats until labels reach sufficient confidence or a maximum number of views, forming the basis for all experiments.

## 5 METHODOLOGY

In this section, we present our algorithm for LLM-guided multi-view object labeling. As described in Section 4 , our images are first processed with the *DetectAndSegment* function, which we implement in a fully open-vocabulary manner. We use YOLOE (Wang et al., 2025) in a prompt-free setup for object detection, which produces candidate labels across thousands of object classes. OpenCLIP (Cherti et al., 2023) is employed for feature extraction, and SAM2 (Ravi et al., 2024) for segmentation. The resulting detections are spatially merged with the *MergeObservations* function, yielding a structured map of objects, each associated with detection images and embeddings. Next, the map is passed to LADR's reasoning step, *RefineAndPropose*, which operates as an inner-loop implemented with GPT-4V (OpenAI, 2024). To clarify the contribution of each component, we introduce the method incrementally through two ablated versions before presenting our complete algorithm.

### 5.1 LLM-RANDOM: BASIC HYPOTHESIS PROPOSAL AND KILLING

The first ablated version, **LLM-Random**, introduces the fundamental hypothesis-proposal and iterative image removal procedure. In multi-view labeling, the evolving set of detection images at each iteration often contains a mix of highly informative canonical views, ambiguous perspectives, and redundant observations. Presenting all available images to the LLM simultaneously is problematic: it risks pushing the model toward a generic, lowest-common-denominator label, substantially increases computational cost, and may even exceed the LLM's context window. A possible workaround is to tile multiple views into a single composite image, but this forces downsampling that discards fine-grained details, an issue that becomes increasingly severe as the number of detections grows. To address these challenges, we adopt an iterative inner loop that samples a small subset of images to form a hypothesis and then prunes away views that conflict with it. The LLM-Random variant implements this process using the simplest possible sampling strategy: uniform random selection. The workflow of the algorithm is illustrated in Figure 1. At each iteration:

1. Randomly sample a minimal set $\mathcal{I}_{sample}$ from the detection images (e.g., $N = 2$ from $\mathcal{I}$).
2. $\mathcal{I}_{sample}$ are fed to the LLM, which is queried to perform three tasks:
    - Propose a label $\mathcal{M}_{\text{refined}}$ based on $\mathcal{I}_{sample}$ and return confidence, with the LLM prompted to report confidence only if the label is clear from all sampled images.
    - Identify which image $I_{kill}$ is the least descriptive of the current label from $\mathcal{I}_{sample}$.
    - Suggest the next best view $P_{next}$ to capture, based on the provided camera angles $\mathcal{P}$.
3. Remove $I_{kill}$ from the detection set.

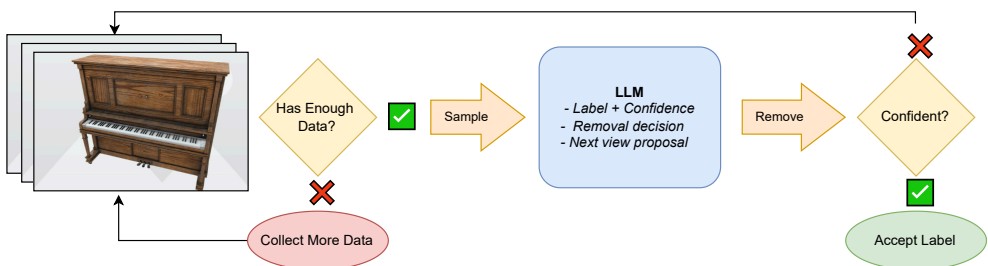

Figure 1: **Workflow of the LLM-guided multi-view labeling algorithm**. The system iteratively refines the semantic map and proposes the next viewpoint to be collected.

The process repeats until either (1) the LLM reports confidence in its label, or (2) the detection set $\mathcal{I}$ has been reduced to fewer than $N$ images (3) a maximum number of iterations is reached; in which case the algorithm returns the refined map $\mathcal{M}_{\text{refined}}$ along with the next proposed viewpoint $P_{\text{next}}$. The LLM prompt used for this algorithm is provided in Appendix A.8.

## 5.2 LLM-SAMPLING: CLIP-GUIDED SELECTION AND CONFIDENCE

The **LLM-Sampling** algorithm follows similar structure as LLM-Random, but improves upon it by leveraging image embeddings provided by a contrastive VLM (eg. CLIP, Cherti et al. (2023)) for both image selection and confidence assessment; we refer to these embeddings as *CLIP features* in the remainder of the text. In the following descriptions, we illustrate the method using two sampled images per iteration for clarity. In practice, this generalizes to two subsets of images, sampled in the same way as individual images. We provide a sketch of an iteration in Figure 2.

**Sampling** Instead of randomly selecting images, the algorithm identifies two images $I_{\text{rep}}, I_{\text{amb}} \subset \mathcal{I}$ based on their cosine similarity of CLIP features relative to the current label hypothesis: the closest (most representative) and the farthest (potentially ambiguous) image. The initial hypothesis can be set using the most common detection label (eg. YOLO detections) or any starting label. The selected images $I_{\text{rep}}$ and $I_{\text{amb}}$ are then fed to the LLM to generate a new label hypothesis $\mathcal{M}_{\text{refined}}$ and propose the next best view $P_{\text{next}}$. This sampling procedure balances *exploitation* (focusing on the most representative view) with *exploration* (including a diverse, informative view).

**Confidence Computation and Removal** A global object representation is computed by averaging CLIP features across all current images in $\mathcal{I}$. Cosine similarity between this global feature and the LLM label embedding $\mathcal{M}_{\text{refined}}$ provides a confidence score for the proposed label. Similarities are computed between $\mathcal{M}_{\text{refined}}$ and the sampled detections $I_{rep}, I_{amb}$. If removing the less similar detection improves global confidence, it is discarded from $\mathcal{I}$. To determine whether the current label is reliable enough to be accepted or if further iterations with new images are required, a confidence threshold is applied (see Appendix A.1 for its calibration).

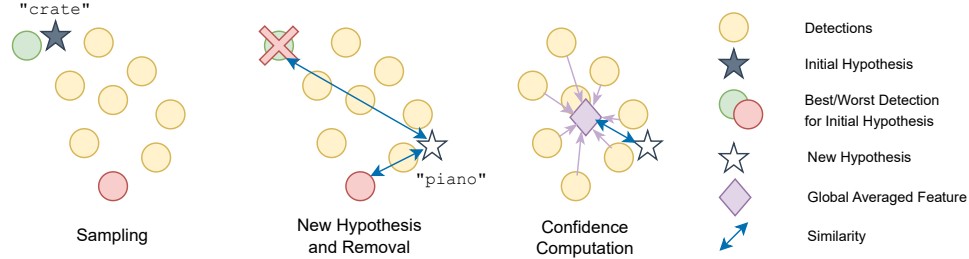

Figure 2: **Visualization of the LLM-Sampling algorithm**: (left) two images are selected based on feature distance from current hypothesis, (middle) a new label hypothesis is generated from the two images, and the less similar detection is removed, (right) a global averaged feature and global confidence are computed.

**Recovering the Final Hypothesis via Caching**   In some cases, the LLM may generate an accurate label early on, but it cannot yet be accepted due to insufficient supporting evidence. The CLIP-similarity-based confidence computation allows for the re-evaluation of previously generated hypotheses. Specifically, at each iteration, the current label hypothesis and its associated CLIP embedding are stored in a cache. The cache enables the algorithm to efficiently compute similarity scores (i.e., confidences) between past hypotheses and the current detections without re-querying the LLM. When new images are introduced or when the hypothesis-proposal loop concludes, the most confident hypothesis is retrieved from the cache. Empirically, this mechanism reduces noise from LLM hallucinations, prevents sudden label shifts, and improves convergence consistency.

**Advantages**   By selecting images uing CLIP feature distances, the LLM receives more informative samples instead of random subsets, reducing redundancy and improving efficiency. At the same time, confidence derived from CLIP features offers a more reliable measure than the LLM's self-reported confidence in LLM-Random. Hypothesis caching ensures that all candidate labels are tracked and reconsidered efficiently, allowing the system to leverage past insights without repeated LLM calls. Together, these strategies efficiently leverage the generative capabilities of LLMs and the contrastive capabilities of CLIP models.

## 5.3   LLM-POLYGON: SPATIALLY GROUNDED REFINEMENT

The complete algorithm, **LLM-Polygon**, extends LLM-Sampling by incorporating spatial grounding into the label refinement process. This addition allows the algorithm to reason about coverage of the object's geometry and to prioritize views that reduce semantic uncertainty, see Figure 3.

**Spatial Assignment**   A right-prism polygon (or an icosahedron) is constructed around the object to approximate its spatial extent. Each detection is associated with the polygon faces it observes, determined by projecting camera rays on the polygon faces. This partition grounds the detections into spatially meaningful subsets and prevents over-representation of individual sides.

**Per-Face Confidence**   For each polygon face, CLIP features of the associated detections are averaged to form a local feature representation. Unobserved faces are assigned an *uncertainty weight*, a hyperparameter that trades off exploration and exploitation: lower uncertainty weights promote taking additional views, while higher values enable faster convergence by downweighting unseen sides (see Appendix A.1 for a calibration guide). Global confidence is then computed as the average similarity between the current label hypothesis and the per-face features. Faces with only removed images are not taken into account when computing the global average confidence.

**Iterative Refinement**   Label proposal and image pruning proceed as in LLM-Sampling, but the next viewpoint is chosen using spatial confidence and coverage. Specifically, $P_{next}$ is selected as the face whose neighboring faces exhibit the largest confidence difference, with priority given to previously unseen sides. This active mechanism directs exploration toward underrepresented object regions while reducing the influence of redundant or uninformative views.

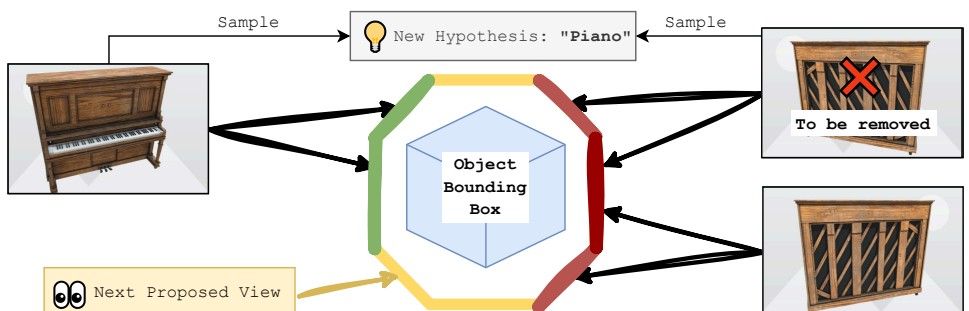

Figure 3: **Illustration of LLM-Polygon**: Object detections are spatially grounded to polygon faces. Per-face confidence is computed based on CLIP features: green sides correspond to high visual similarity to the current label ("piano"), red sides indicate low similarity, and yellow sides represent unseen faces. The next camera viewpoint is selected to reduce uncertainty, prioritizing unseen faces.

## 6 EXPERIMENTS

We evaluate our proposed method, LADR, against several baseline algorithms in both single-object and multi-object settings. The experiments are designed to assess each method's ability to infer object semantic labels accurately under controlled multi-view scenarios. On single-object scenes, we demonstrate that reasoning with a large language model is crucial for consistent 3D object detection and that active view selection greatly improves both sample efficiency and stability. In addition to single-object experiments, we evaluate all methods on multi-object scenes, a realistic setting for robot exploration. Here, we isolate the impact of our label generation mechanism by using off-the-shelf exploration policies rather than proposing next-best views. This setup highlights that our representation still improves multi-object labeling as an offline refinement process. Full details of all hyperparameters used are provided in Appendix A.2.

### 6.1 BASELINES

We compare methods that rely solely on YOLO detections, CLIP embeddings, or LLM reasoning with LADR, which leverages multi-view aggregation, spatial grounding, and confidence-based label selection. Here, LADR refers to our three algorithms: **LLM-Random, LLM-Sampling, and LLM-Polygon**. Apart from these, only the **LLM-Angle** baseline explicitly proposes the next best view; for all other baselines, random view sampling is used when not otherwise specified.

**'YOLO'**: uses the most common label (provided by YOLO detections) as the final label. This is the aggregation policy in ConceptFusion (Jatavallabhula et al., 2023).

**'CLIP'**: takes the average of the CLIP embeddings of all images and compares it to an extensive list of CLIP-embedded labels of the RAM class list (Zhang et al., 2023). The final label is the one with the highest cosine similarity to the average embedding.

**'LLM-Label'**: analyzes the YOLO-generated detection labels using an LLM. The LLM reasons over the set of labels, their frequencies, and possible semantic relationships to infer the most plausible final label. No visual data is used, only text-based outputs from the object detector.

**'LLM-Tiled'**: creates a single composite image by arranging all input images on a single frame with optimal tiling. This layout is then analyzed by a large language model with vision capabilities to produce the final label. The spatial layout enables the model to reason about object appearance from all sides simultaneously. (We provide example in Appendix A.7).

**'LLM-Angle'**: creates a single composite image by arranging all input images around a circle based on their relative positions to the object. This panoramic-style layout is then analyzed by a large vision model to produce the final label. The spatial layout enables the model to reason about object appearance from all sides simultaneously. Unlike the other baselines, the LLM also provides the angle for the next best view to take an image from. (We provide example in Appendix A.7).

### 6.2 DATASET

We evaluate our methods on a single-object dataset, a subset of the *OmniObjects3D* (Wu et al., 2023) dataset of annotated 3D object models. These objects are rendered in NVIDIA Isaac Sim under controlled conditions to generate multi-view image sequences. We focus on five object classes: backpack, cup, cabinet, sofa, and suitcase. For each class, we include five distinct instances, several of which are deliberately misleading in appearance (e.g., a mug shaped like a cartoon character) to test the robustness of semantic labeling methods.

We also constructed a multi-object dataset in the same simulation environment. These scenes contain multiple objects arranged in varied environments, including SimpleRoom, Commercial, Industrial, Residential, and Vegetation, providing more complex scenarios with occlusions. For simplicity, these are single-room scenes, and the objects were selected to include both easy and more challenging cases to label.

Each object in the datasets is annotated with both its class name and a concise descriptive phrase, for example a chair labeled as *chair* with the description *wooden dining chair with a cushioned seat*. We provide examples for both datasets in Appendix A.4.

## 6.3 EXPERIMENTAL SETUP

**Single-Object Experiment**  We evaluate all baselines on single-object datasets under controlled conditions, simulating a robotic-agent use case. For each object, we provide two initial views captured at a fixed height against a white background. Each algorithm is tested with three different random seeds per object instance, corresponding to three distinct pairs of these initial views. We set a maximum budget of five additional views.

**Multi-Object Experiment**  The multi-object setting evaluates LADR in more complex environments with multiple objects and occlusions, geared towards more realistic robotic-agent settings. We restrict experiments to single-room scenarios and define robot positions as the unit of exploration. At each position, the robot acquires $k = 8$ uniformly spaced RGB-D images. Instead of proposing next-best views, we employ global exploration policies over a 2D occupancy grid to determine the robot's next position. We investigate three strategies: **random**, selecting a free cell uniformly at random; **medial axis** (van der Walt et al., 2014), sampling a random point on the medial axis of the free space using `scikit-image`; and **frontier-based** (Yamauchi, 1997), prioritizing frontier cells or selecting a random free cell if no frontiers remain. Exploration starts from a randomly chosen initial position and continues until the maximum budget of three additional positions is reached.

## 6.4 EVALUATION METRICS

To assess labeling performance, the predicted labels are compared against both the ground-truth object class names and longer, descriptive phrases for each object (e.g., "yellow cartoon character-shaped mug"). As LADR operates in a fully open-vocabulary setting, direct comparison with ground-truth labels is not sufficient: the LLM may propose synonyms of the annotated class, which should be accepted as correct. Empirically, we found that the CLIP model used for image–text similarity is overly sensitive to lexical variation (e.g., number of words in a label), leading to unreliable synonym matching. Instead, we employ a Sentence Transformer (Reimers & Gurevych, 2019) model to evaluate label equivalence. The final similarity score for each prediction is defined as the maximum of the similarity to the class name and the similarity to the description, capturing both category-level and instance-level alignment. To evaluate success rates rather than raw similarities, we adopt the similarity value 0.5 as the threshold for label correctness (based on preliminary experiments; see Appendix A.3), while also considering thresholds of 0.3, 0.7, and 0.9.

To evaluate detections in the multi-object setting, we establish one-to-one matches between ground-truth objects and predicted detections from the global map. Matching is based on a semantic-spatial similarity score, defined as a weighted sum of label similarity and spatial overlap between ground-truth and predicted bounding boxes. Once matches are established, evaluation metrics follow the same procedure as in the single-object setting, ensuring comparability.

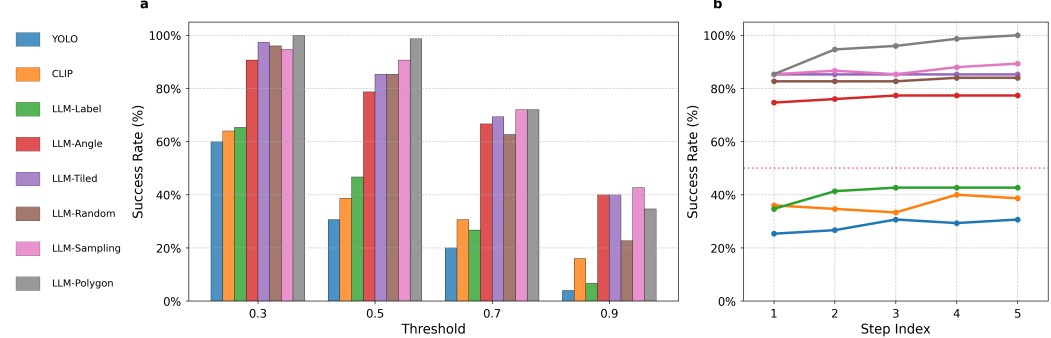

Figure 4: **Single-Object Experiment Results** (a) Averaged success rates across different success thresholds for each algorithm. (b) Evolution of success rates over data collection steps for each algorithm, using 0.5 as the threshold.

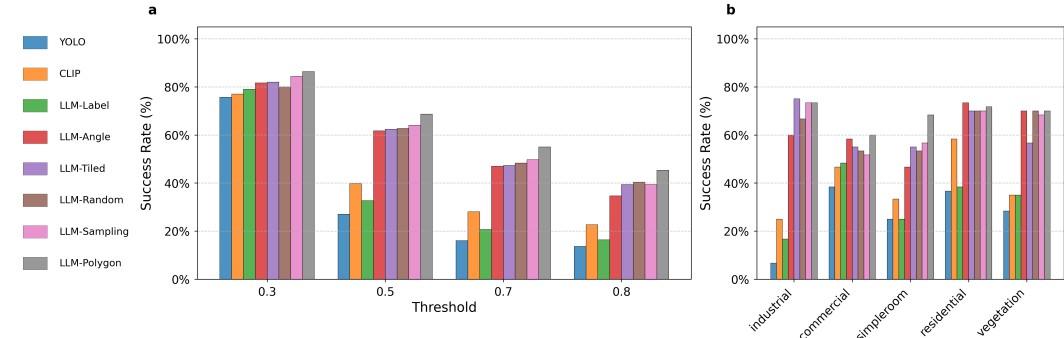

Figure 5: **Multi-Object Experiment Results** (a) Averaged success rates across different success thresholds. (b) Averaged success rates across scenes, using 0.5 as the threshold.

### 6.5 SUMMARY OF FINDINGS

We provide our results for the single- and multi-object cases in Figures 4 and 5, respectively. We provide detailed results, including per-object examples for each setting in Appendices A.5 and A.6. Figure 4a shows the averaged success rates based on different success thresholds, and Figure 4b shows how success rates evolve over the data collection steps with 0.5 as the threshold.

Similar trends are observed for single- and multi-object cases. The first observation is that they show over 40% improvements, respectively, compared to ubiquitous fusion methods using YOLO, and CLIP. **YOLO** and **LLM-Label** rely solely on YOLOE predictions, resulting in consistently low success rates. This is likely due to their lack of multi-view image-based reasoning. Notably, LLM reasoning alone offers little improvement over simply taking the most frequent YOLO label. **CLIP** performs comparably to YOLO, but struggles with the vast label set and the ambiguity introduced by averaging embeddings across views, often leading to confused predictions. **LLM-Tiled** achieves higher success rates by leveraging all views simultaneously. However, its accuracy lags behind LADRs, suggesting that the tiled representation either loses fine-grained detail or introduces structural incoherence that limits reasoning. **LLM-Angle** adds structural consistency by composing views in an ordered layout, yet provides no improvement over LLM-Tiled. This indicates that the performance gap is more likely due to loss of visual detail than to layout incoherence. **LLM-Random** and **LLM-Sampling** analyze images in greater detail, leading to stronger descriptive accuracy. However, LLM-Random often declares detections prematurely, and LLM-Sampling cannot fully mitigate this instability despite its confidence-based pruning. Finally, **LLM-Polygon** achieves the best overall performance, with near-perfect success at a 0.5 threshold. By combining detailed reasoning with active exploration of informative views and consistency across unseen sides, it avoids the pitfalls of LLM-Only and LLM-Sampling. Figure 4 /b shows how active exploration of unseen sides leads to success rate improvement. LADR's combination of smart view sampling, confidence computation, and spatial grounding is key to outperform approaches that naively provide multiple images to the LLM, as in LLM-Tile and LLM-Angle.

### 7 CONCLUSION

Our contributions in this work center on a scalable framework for iterative sampling with LLM-guided active refinement and exploration in open-vocabulary 3D object detection. By integrating the generative reasoning of LLMs with the quantitative similarity assessment of contrastive VLMs, our approach substantially improves label consistency, establishing a foundation for future research in robust and efficient 3D perception. The method also serves as a drop-in extension to existing object detection pipelines, allowing zero-shot re-evaluation of detections.

Despite these advantages, the methods presented require multiple inner-loop queries, which increases computational cost. This limitation could be mitigated through batch sampling strategies, or pruning multiple detections simultaneously, as well as by employing more efficient vision-language models, e.g., FastVLM (Vasu et al., 2025), to enable inference on resource-constrained hardware.

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

# A APPENDIX

## A.1 HYPERPARAMETER CALIBRATION

We calibrate the two key hyperparameters, the confidence threshold and uncertainty weight, using a held-out subset of 5k COCO images. Matching (image–correct label) and non-matching (image–incorrect label) pairs are used to compute CLIP similarities, with distributions estimated via kernel density estimation (KDE). As shown in Figure 6, the distributions are well-separated (Cohen's $d = 5.06$).

The confidence threshold is set to the mean similarity of matching pairs ($\mu_+ = 0.311$, rounded to 0.32), and the uncertainty weight is set to the intersection of the distributions (rounded to 0.2), corresponding to the Bayes-optimal decision boundary.

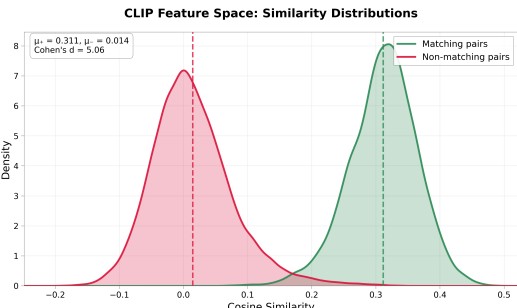

Figure 6: **CLIP Feature Space: Similarity Distributions.** The plot shows the kernel density estimates (KDE) of cosine similarity distributions for matching (image–correct label) and non-matching (image–incorrect label) pairs within the CLIP feature space, using a held-out subset of COCO. The green distribution represents matching pairs, and its mean ($\mu_+ = 0.311$) is used as the **confidence threshold** (dashed green line), rounded to 0.32. This ensures that accepted labels correspond to in-distribution confidence levels. The red distribution represents non-matching pairs. The intersection point of the two distributions, which corresponds to the Bayes-optimal decision boundary, is used to set the **uncertainty weight**, rounded to 0.2.

## A.2 HYPERPARAMETERS

We present the hyperparameters used in our experiments at each step of the workflow. All images are 1920x1080 to preserve high-level detail.

***DetectAndSegment*** YOLOE (Wang et al., 2025) in prompt-free mode is used to generate object bounding boxes. Detections with fewer than 5,000 pixels or confidence below 0.15 are discarded. Features are extracted using OpenCLIP (Cherti et al., 2023) ViT-H-14 with the "laion2b_s32b_b79k" weights, and segmentation is performed with Segment Anything 2 (Ravi et al., 2024).

***MergeObservations*** In single-object experiments, all detections are assumed to correspond to the same object and are merged. In multi-object experiments, observations are merged based on semantic (visual and textual) and spatial similarity. Visual semantic similarity is computed as the average CLIP features of detections (excluding removed ones from the RefineAndPropose step) with a threshold of 0.6. Textual similarity is the cosine similarity of CLIP-encoded current object labels, thresholded at 0.25. Spatial similarity is measured via point cloud overlap, with a threshold of 0.1 to allow minimal overlap. Objects exceeding all three thresholds are merged.

***RefineAndPropose*** For both experiments, we set the maximum number of inner-loop iterations between data collection steps to 3, the confidence threshold to 0.32, the uncertainty weight to 0.2, and the number of polygon faces in the spatial partitioning algorithm to 8.

## A.3 EVALUATION

### A.3.1 SETTING THE SUCCESS THRESHOLD

As LADR is fully open-vocabulary, direct comparison with ground-truth labels is insufficient: the LLM may propose synonyms, which should be accepted. Since CLIP is sensitive to lexical variations, we use a Sentence Transformer (Reimers & Gurevych, 2019) to evaluate label equivalence. The final similarity for each prediction is the maximum of its similarity to the class name or description. To convert similarities into success rates, we construct a small set of synonym and non-synonym pairs, compute their similarities in the Sentence Transformer feature space, and visualize the distributions using kernel density estimation (KDE). The results show clear separation: while matching pairs can occasionally fall below 0.5, non-matching pairs never exceed 0.5. Based on this, we adopt 0.5 as the default threshold for evaluating label correctness.

To provide a more nuanced view, we also evaluate success rates at multiple thresholds:

- **0.3:** Almost all word pairs are detected as synonyms, including weakly related or contextually distant ones.
- **0.5:** Serves as a baseline, capturing meaningful synonyms while avoiding unrelated pairs.
- **0.7:** Mostly multi-word phrases with strong semantic alignment; loosely related pairs are excluded.
- **0.9:** Captures nearly identical or identical pairs, useful for exact matches.

By reporting success rates at these thresholds, we provide a more detailed picture of the model's behavior across varying levels of semantic similarity, from broad synonym detection to nearly exact matches.

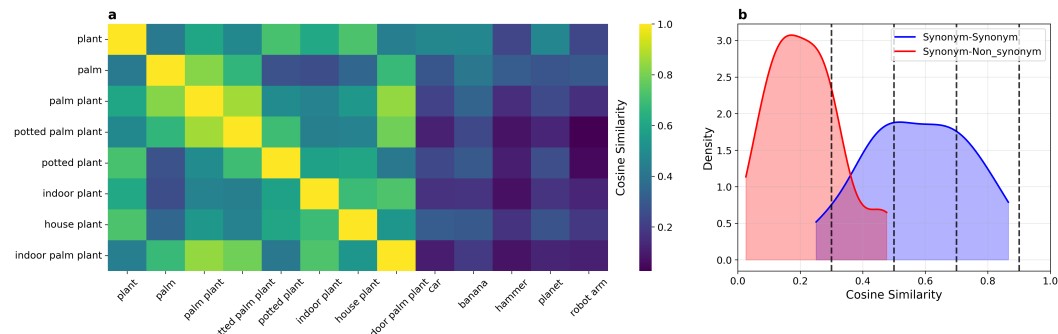

Figure 7: **Synonym Distance Analysis (a)** A cosine similarity heatmap for the word 'plant' and a set of related terms. The diagonal entries show high similarity for synonyms and near-synonyms (e.g., 'plant', 'potted plant', 'indoor plant'). Non-synonyms (e.g., 'car', 'banana', 'planet') exhibit low similarity. **(b)** Kernel density estimate (KDE) plots of cosine similarity distributions for synonym (blue) and non-synonym (red) pairs. The distributions show a clear separation, with a default threshold of 0.5 effectively distinguishing between the two. The dashed lines indicate various thresholds (0.3, 0.5, 0.7, and 0.9) used to evaluate the model's performance at different levels of semantic similarity, from broad synonym detection to near-exact matches.

### A.3.2 MATCHING DETECTIONS TO GROUND TRUTH

To evaluate multi-object detections, we assign each ground-truth object to the best-matching prediction based on a semantic-spatial similarity score, computed as a weighted combination of label similarity and spatial overlap. Only matches with similarity above 0.1 are considered; lower values count as unsuccessful detections. Among eligible matches, the final assignment uses a bias-adjusted aggregation with phys_bias = 0.2 to select the best match.

### A.4 DATASETS

#### A.4.1 SINGLE-OBJECT DATASET

The single-object dataset comprises five instances for each of the five selected object classes from the OmniObjects3D dataset (Wu et al., 2023). These instances are used to generate multi-view image sequences, with representative examples shown in Figure 8.

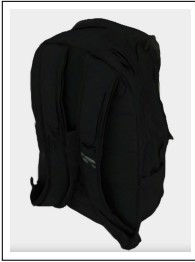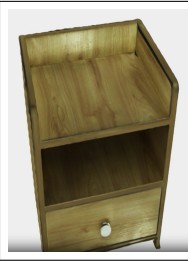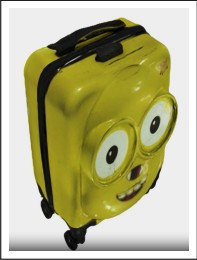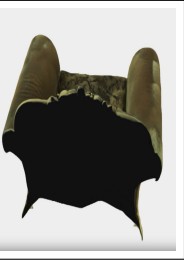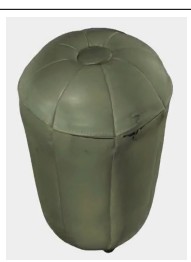

Figure 8: Examples of five object instances from the single-object experiments.

#### A.4.2 MULTI-OBJECT DATASET

The multi-object dataset consists of custom 3D scenes created in NVIDIA Isaac Sim and manually labeled by the authors. To demonstrate the flexibility of our approach, we designed diverse environments using simulator-provided asset packs. The included room types are:

- **SimpleRoom:** open indoor spaces with a mix of miscellaneous objects,
- **Residential:** home-like settings with rug, chairs, and decorative items,
- **Commercial:** office area with a counter, a coffee-table and a storage unit,
- **Industrial:** warehouse-inspired space with shelving, crates, and utility equipment,
- **Vegetation:** outdoor theme featuring plants, trees, and garden elements.

Each scene contains multiple objects of interest, with dense arrangements to test robustness under occlusions, see Figure 9.

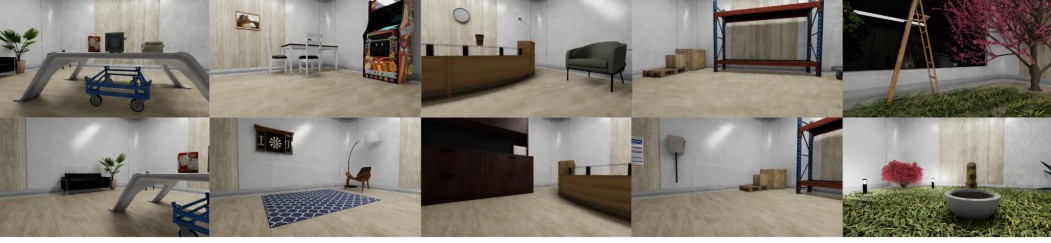

| SimpleRoom | Residential | Commercial | Industrial | Vegetation |
|---|---|---|---|---|

Figure 9: Five room scenes used in the multi-object experiments

#### A.4.3 EXPERIMENT CONFIGURATION

In the single-object experiments, we average over 75 detections (5 classes × 5 instances × 3 seeds), starting with two initial views and allowing a budget of five additional views. In the multi-object setting, we average over 300 detections (5 scenes × 10 objects × 3 exploration policies × 2 seeds). At each position, eight new images are captured, beginning from a single initial position with a budget limit of three additional positions.

## A.5 SINGLE-OBJECT EXPERIMENT RESULTS

### A.5.1 DETAILED RESULTS

| Algorithm | Class Sim | Desc Sim | Best Sim | Avg Sim | Succ@0.3 | Succ@0.5 | Succ@0.7 | Succ@0.9 | Avg Tokens |
|---|---|---|---|---|---|---|---|---|---|
| YOLO | 0.41 ± 0.25 | 0.31 ± 0.22 | 0.43 ± 0.26 | 0.36 ± 0.23 | 0.60 | 0.31 | 0.20 | 0.04 | 0 |
| CLIP | 0.51 ± 0.29 | 0.35 ± 0.20 | 0.52 ± 0.28 | 0.43 ± 0.23 | 0.64 | 0.39 | 0.31 | 0.16 | 0 |
| LLM-Label | 0.49 ± 0.28 | 0.38 ± 0.21 | 0.50 ± 0.28 | 0.42 ± 0.24 | 0.65 | 0.47 | 0.27 | 0.07 | 237 |
| LLM-Angle | 0.68 ± 0.31 | 0.59 ± 0.21 | 0.74 ± 0.27 | 0.64 ± 0.23 | 0.91 | 0.79 | 0.67 | 0.40 | 1575 |
| LLM-Tiled | 0.72 ± 0.27 | 0.62 ± 0.17 | 0.78 ± 0.23 | 0.67 ± 0.19 | 0.97 | 0.85 | 0.69 | 0.40 | 1008 |
| LLM-Random | 0.66 ± 0.26 | 0.62 ± 0.17 | 0.73 ± 0.21 | 0.64 ± 0.19 | 0.96 | 0.85 | 0.63 | 0.23 | 2182 |
| LLM-Sampling | 0.71 ± 0.30 | 0.63 ± 0.17 | 0.79 ± 0.23 | 0.67 ± 0.20 | 0.95 | 0.91 | 0.72 | 0.43 | 16115 |
| LLM-Polygon | 0.73 ± 0.24 | 0.66 ± 0.14 | 0.80 ± 0.17 | 0.69 ± 0.15 | 1.00 | 0.99 | 0.72 | 0.35 | 22176 |

Table 2: Detailed evaluation results for different algorithms. Similarity metrics are reported as mean ± standard deviation, followed by success rates at various thresholds and average LLM tokens used.

### A.5.2 SAMPLE RESULTS

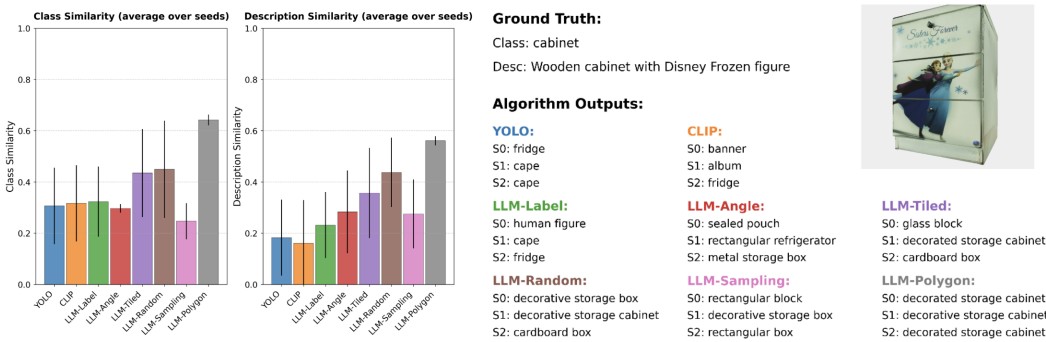

Figure 10: A per-object example showing algorithm performance. The bar charts on the left present class and description similarity, averaged over the seeds, while the right provides a qualitative example for an object in the 'cabinet' category from the single-object dataset. This example highlights that the generic 'cabinet' label is not sufficiently descriptive for this particular object.

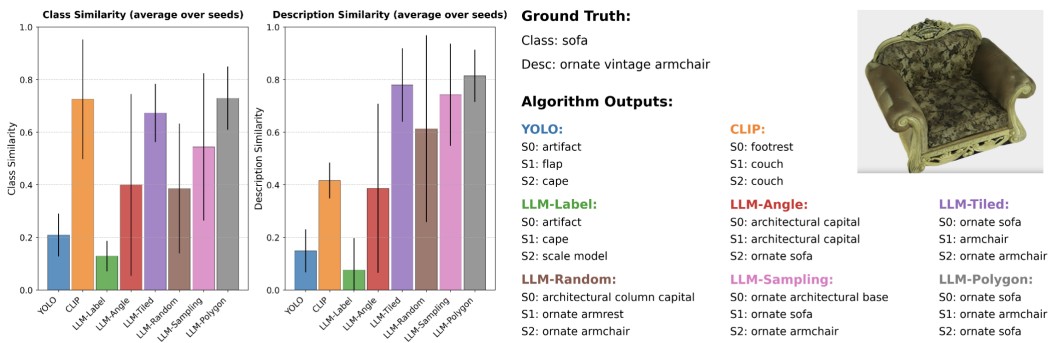

Figure 11: A per-object example showing algorithm performance. The bar charts on the left present class and description similarity, averaged over the seeds, while the right provides a qualitative example for an object in the "sofa" category in the single-object dataset.

## A.6 Multi-Object Experiment Results

### A.6.1 Detailed Results

| Algorithm | Class Sim | Desc Sim | Best Sim | Avg Sim | Succ@0.3 | Succ@0.5 | Succ@0.7 | Succ@0.9 | Avg Tokens |
|---|---|---|---|---|---|---|---|---|---|
| YOLO | $0.45 \pm 0.24$ | $0.34 \pm 0.18$ | $0.46 \pm 0.24$ | $0.39 \pm 0.21$ | 0.76 | 0.27 | 0.16 | 0.09 | 0 |
| CLIP | $0.51 \pm 0.28$ | $0.40 \pm 0.23$ | $0.52 \pm 0.28$ | $0.45 \pm 0.24$ | 0.77 | 0.40 | 0.28 | 0.16 | 0 |
| LLM-Label | $0.48 \pm 0.26$ | $0.38 \pm 0.20$ | $0.49 \pm 0.25$ | $0.43 \pm 0.22$ | 0.79 | 0.33 | 0.21 | 0.12 | 2965 |
| LLM-Angle | $0.56 \pm 0.28$ | $0.54 \pm 0.28$ | $0.63 \pm 0.30$ | $0.55 \pm 0.26$ | 0.82 | 0.62 | 0.47 | 0.26 | 8350 |
| LLM-Tiled | $0.57 \pm 0.27$ | $0.56 \pm 0.28$ | $0.64 \pm 0.30$ | $0.57 \pm 0.26$ | 0.82 | 0.62 | 0.47 | 0.25 | 6412 |
| LLM-Random | $0.56 \pm 0.27$ | $0.57 \pm 0.28$ | $0.64 \pm 0.29$ | $0.57 \pm 0.26$ | 0.80 | 0.63 | 0.48 | 0.26 | 12496 |
| LLM-Sampling | $0.59 \pm 0.28$ | $0.56 \pm 0.28$ | $0.65 \pm 0.29$ | $0.58 \pm 0.26$ | 0.84 | 0.64 | 0.50 | 0.29 | 14278 |
| LLM-Polygon | $0.59 \pm 0.26$ | $0.61 \pm 0.28$ | $0.67 \pm 0.28$ | $0.60 \pm 0.25$ | 0.86 | 0.69 | 0.55 | 0.27 | 17633 |

Table 3: Detailed evaluation results for different algorithms. Similarity metrics are reported as mean ± standard deviation, followed by success rates at various thresholds and average LLM tokens used.

### A.6.2 Sample Results

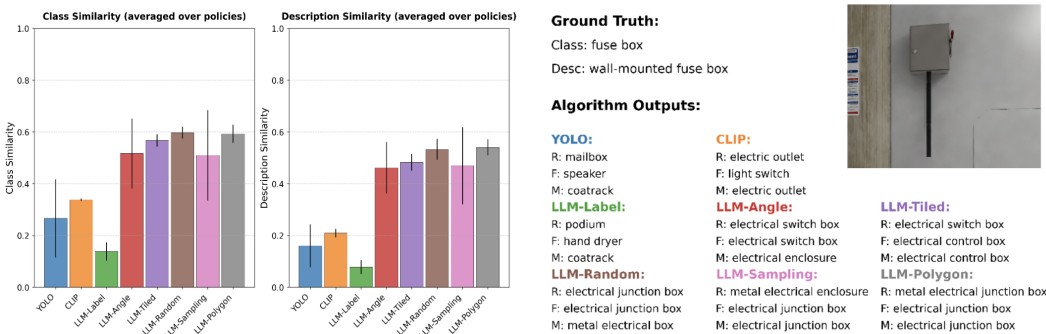

Figure 12: A per-object example showing algorithm performance. The bar charts on the left present class and description similarity, averaged over the exploration policies, while the right provides a qualitative example for an object with "fuse box" as the ground truth label in the multi-object dataset.

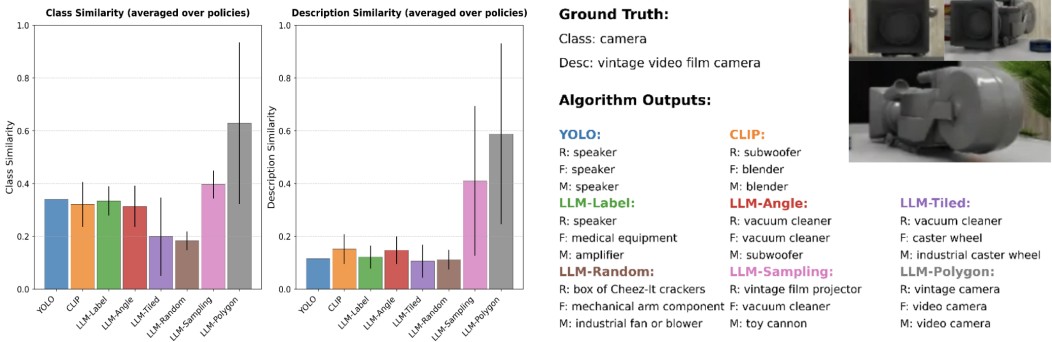

Figure 13: A per-object example showing algorithm performance. The bar charts on the left present class and description similarity, averaged over the exploration policies, while the right provides a qualitative example for an object with "camera" as the ground truth label in the multi-object dataset.

## A.7 LLM IMAGERY INPUT DATA

We provide examples of the LLM-Angle, and LLM-Tile in Figure 14.

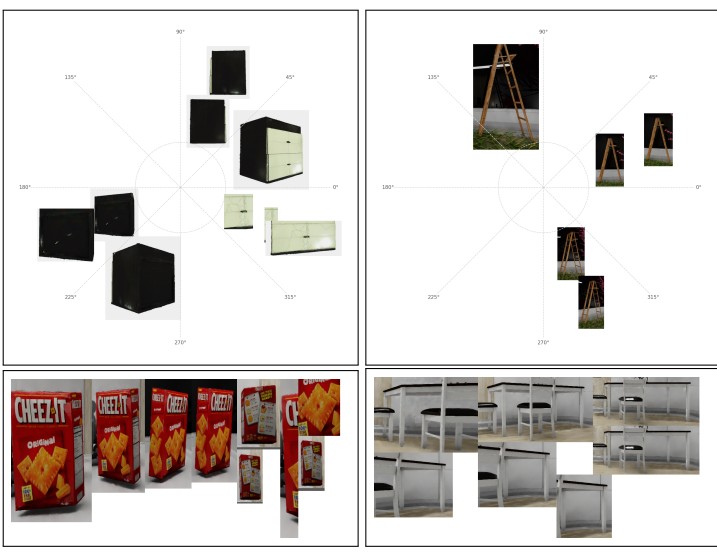

Figure 14: Top: input for LLM-Angle. Bottom: Input for LLM-Tile.

## A.8 LLM PROMPT

**Example of the prompt provided to the LLM for object labeling.** We provide the prompt used for LLM-Random. The prompts for other LADR algorithms are largely similar, with a few differences: neither LLM-Sampling nor LLM-Polygon requests a confidence score or the more descriptive view, and LLM-Polygon also does not request the next-best-view suggestion.

```
You will receive two images of the same object taken from (different) viewpoints, along with
↪  the angles (in degrees) from which they were captured. Analyze both images together
↪  considering their angles and return a single JSON object with these fields:

confident: true or false, indicating whether you are fully confident in the objects class
↪  based on the two views.
label: a brief class name of the object.
description: a clear, detailed description of the object for CLIP encoding. Focus on visually
↪  distinctive features (shape, material, color, texture, patterns) observable in at least
↪  one image.
next_best_angle: an integer in the range [-180, 180] suggesting the single most informative
↪  angle for revealing any ambiguous or missing features.
more_descriptive: either "left" or "right", indicating which image shows features most
↪  representative of the labeled class.
explanation: a short rationale covering:
    - why you set confident to true or false;
    - how you chose label and description;
    - why the proposed next_best_angle will improve clarity;
    - why the chosen image (\left" or \right") is more descriptive.

Guidelines:

Focus on the Main Object.
Each image is a crop around the objects bounding box, and the object fills most of the frame.
↪  Ignore background elements or smaller occluded items.

Combine Both Views and Angles.
Use both images and their provided angles to form a complete understanding. One view may
↪  reveal overall shape, while the other shows texture or details. Identify any remaining
↪  ambiguity or blind spots when choosing your next_best_angle.

Avoid Misidentifying from Partial Views.
If one image shows only a fragment (e.g., a handle), defer to the other image for overall
↪  class identification. Do not let a partial segment mislead your label.
```

```
Highlight Distinctive Features.
Describe only the most visually salient characteristics clearly visible in at least one image.
↪  Write in plain, factual language similar to alt-text or OpenCLIP-style captions.

Assess Confidence.
Set confident to true only if both images clearly support the same object class. If you
↪  suspect the label might change from another viewpoint or if one view is ambiguous, set
↪  confident to false and propose a next_best_angle that would resolve that ambiguity.

Determine \More Descriptive" View.
Compare the two images (left vs. right). Whichever one shows features most representative of
↪  the labeled classwhether by revealing overall shape, distinctive markings, or full
↪  extentshould be marked in more_descriptive. If both show equal detail, choose the one
↪  closest to the objects canonical appearance.

Next-Best-View Proposal.
Recommend a single integer angle in [-180, 180] that would most improve clarity of class or
↪  reveal missing features. Base your suggestion on the two given angles. For example, if the
↪  provided images are at 45 (left) and 60 (right), proposing 0 might reveal the front;
↪  proposing 90 might reveal the opposite side.

Be Precise and Concise.
Write factually. Avoid speculation beyond what the two views suggest. Do not use generic class
↪  labels unsupported by the images.

Output Format
Return exactly one JSON object, for example:
{
  "confident": false,
  "label": "ceramic vase",
  "description": "a rounded ceramic vase with a narrow neck and blue floral patterns on a
  ↪  white background",
  "next_best_angle": 0,
  "more_descriptive": "right",
  "explanation": "The right image clearly shows the floral pattern and vase shape, but the
  ↪  left image only reveals the neck. Because the base is not visible from either 30 or 45,
  ↪  a 0 angle would show the full body and confirm the class."
}

Ensure that your JSON is valid, that all fields are present with the correct types, and that
↪  your response is accurate, well-structured, and concise.
Return only the raw JSON object.
```

