# OpenReview forum: "Consistent 3D Object Detection with Active LLM Reasoning"
_ICLR.cc/2026/Conference — ICLR 2026 Conference Withdrawn Submission_

### Official Review · Reviewer_RgHZ · 2025-10-29

**Soundness:** 2
**Presentation:** 2
**Contribution:** 1
**Rating:** 2
**Confidence:** 4

**Summary:**

The paper proposes LADR, a method that leverages LLMs to refine multi-view 3D object labels by iteratively selecting informative views and prompting the LLM to generate or revise semantic labels. It claims improvements over baseline fusion strategies using YOLO and CLIP, particularly in label consistency and sampling efficiency.

**Strengths:**

The high-level idea of using LLMs to resolve viewpoint-induced label inconsistency is intuitive.

**Weaknesses:**

(1) ***Technical Novelty***: The proposed method appears like a sequence of engineering steps rather than a principled algorithm. It is largely a straightforward pipeline stitching together off-the-shelf components (YOLO, CLIP, SAM, GPT-4V) with limited technical novelty.

* **LLM-guided relabeling for 3D consistency**. This is an incremental adaptation of existing multimodal reasoning pipelines. The paper uses off-the-shelf YOLOE, OpenCLIP, SAM2, and GPT-4V to relabel detections. The “relabeling” step is not technically novel, as it amounts to prompting an LLM to reconcile labels across multiple 2D views. There is no new learning objective, model architecture, or mathematical formulation beyond basic feature averaging and LLM prompting.

* **Smart sampling strategy**. While the idea of selecting informative views is framed as “active learning,” the implementation is heuristic. In “LLM-Sampling” (Section 5.2), informative views are chosen by computing cosine distances between CLIP embeddings. This is an established approach in active exploration literature. The variant “LLM-Polygon” (Section 5.3) uses a fixed polygon partition to spatially group detections, but this is again a geometric bookkeeping device, not a new algorithmic principle. The novelty lies mainly in combining these existing heuristics within a single loop, rather than proposing a new active sampling algorithm.

* **Spatial-semantic mapping**. I do not see any new representational or fusion mechanism. The “polygon grounding” (Section 5.3) is a coarse geometric discretization, not a learned or optimized structure.


(2) ***Experimental Validation***: The paper does not include any ablation studies, which are critical to validate its claims. A key missing ablation would be to isolate the contribution of LLM reasoning versus CLIP-only sampling. Other missing experiments include sensitivity to LLM prompt and model choice, statistical robustness, scalability, and runtime analysis.

**Questions:**

Please refer to my comments in the Weaknesses section.

---

### Official Review · Reviewer_DeFN · 2025-10-30

**Soundness:** 2
**Presentation:** 2
**Contribution:** 2
**Rating:** 4
**Confidence:** 4

**Summary:**

The paper introduces LADR, a framework for achieving consistent multi-view 3D object detection in open-vocabulary settings. The method uses large language models to reason about visual inputs from multiple viewpoints, actively sampling informative views and reweighting label hypotheses based on CLIP feature diversity and confidence. Three variants (LLM-Random, LLM-Sampling, and LLM-Polygon) progressively add structured sampling and spatial grounding. Experiments on single-object and multi-object synthetic datasets demonstrate improvements in label consistency and accuracy over other baselines.

**Strengths:**

- The paper presents an interesting direction, i.e., using LLMs not only as labelers but also as agents that reason about multiple viewpoints and guide data collection.

- Both single- and multi-object scenarios are evaluated with quantitative metrics and qualitative examples, suggesting that the proposed idea yields consistent improvements.

- The stepwise introduction of three algorithm variants provides a clear understanding of each component.

**Weaknesses:**

- While the idea of combining LLM reasoning with CLIP features for 3D detection is interesting, much of the pipeline is a straightforward composition of existing components (YOLOE, SAM2, CLIP, GPT-4V) with a small amount of new learning or modeling innovations.

- The method requires multiple inner-loop LLM queries per object and per iteration, which is computationally expensive and likely infeasible for real-time or large-scale applications.

- Baselines such as CLIP or YOLO fusion are quite weak compared to the rich LLM pipeline. Stronger open-vocabulary 3D methods are not adequately compared.

- The success rate metric relies on heuristic thresholds, making the quantitative gains somewhat fragile. Statistical significance or ablations of hyperparameters are missing.

**Questions:**

- Could the authors provide the computational cost (LLM calls, latency, GPU hours) compared to CLIP or other baselines?
- Since the method heavily depends on GPT-4V, how stable are the results across different LLMs or model versions?
- Does the system generalize beyond object naming (e.g., affordance reasoning or relationship labeling), or is it limited to class identification?

---

### Official Review · Reviewer_wymX · 2025-11-01

**Soundness:** 3
**Presentation:** 2
**Contribution:** 2
**Rating:** 2
**Confidence:** 3

**Summary:**

The paper proposes LADR (LLM-guided Active Detection and Reasoning), a system integrating YOLOE, CLIP, and GPT-4V for multi-view 3D object labeling. It introduces three procedural variants (LLM-Random, LLM-Sampling, LLM-Polygon) that progressively incorporate CLIP-based sampling and spatial grounding to improve label consistency across viewpoints. Experiments are conducted in simulation on single- and multi-object datasets, showing numerical gains over YOLO/CLIP baselines.

**Strengths:**

1. The motivation is clear: improving semantic consistency in open-vocabulary 3D detection.

2. The system is well-engineered and described in detail, including clear workflow diagrams and ablation setups.

3. Quantitative improvements are consistent across tasks.

**Weaknesses:**

1. The contribution is primarily procedural integration of existing components (YOLOE + CLIP + GPT-4V + SAM2), with no novel learning formulation or analytical insight. The workflow could be implemented as an engineering prototype rather than a research contribution.

2. Evaluation lacks depth. Experiments are all in simulated environments (Isaac Sim). No real-world data or generalization study. The claimed “40% improvement” is mostly due to poor baselines rather than a strong methodological leap.

3. Despite claiming to be “truly open-vocabulary” and “reasoning-based,” the system fundamentally depends on CLIP’s embedding space and manually designed prompts. The reasoning is not learned or interpretable.

**Questions:**

See Weaknesses.

---

### Note · Authors · 2026-01-09

I have read and agree with the venue's withdrawal policy on behalf of myself and my co-authors.